# Regional anaesthesia practice in public hospitals in Botswana: A cross-sectional study

**Mamo Kassa[1], Farai Madzimbamuto[1], Gaone Kediegite[1], Eugene Tuyishime**[1,2,3]*

**1** Department Anesthesia and Critical Care, University of Botswana, Gaborone, Botswana, **2** Anesthesia, Critical Care, and Emergency Medicine department, University of Rwanda, Kigali, Rwanda, **3** Anesthesia and Perioperative Medicine department, Western University, London, Canada

* tuyishime36@gmail.com

## Abstract

### Introduction

Little is known about the regional anesthesia practice in low resources settings (LRS). The aim of this study was to describe the regional anesthesia capacity, characteristics of regional anesthesia practice, and challenges and solutions of practicing safe regional anesthesia in public hospitals in Botswana.

### Methods

This was a cross-sectional survey of anesthesia providers working in public hospitals in Botswana. A purposive sampling method of public hospitals was used to achieve representation of different hospital levels across Botswana. Paper-based questionnaires were sent to anesthesia providers from selected hospitals. Descriptive statistics were used for analysis.

### Results

**Questionnaires were distributed to 47 selected anesthesia providers from selected hospitals; 38 (80.9%) were returned.** Most participants were nurse anesthetists and medical officers (57.8%). All hospitals perform spinal anesthesia; however, other regional techniques were performed by a small number of participants in one referral hospital. Most hospitals had adequate regional anesthesia drugs and sedation medications, however, most hospitals (except one referral hospital) lacked ultrasound machine and the regional anesthesia kit. The common challenges reported were lack of knowledge and skills, lack of equipment and supplies, and lack of hospital engagement and support. Some solutions were proposed such as regional anesthesia training and engaging the hospital management to get resources.

### Conclusions

The results of this study suggest that spinal anesthesia is the most common regional anesthesia technique performed by anesthesia providers working in public hospitals in Botswana followed by few upper limb blocks. However, most public hospitals lack enough training

**Data Availability Statement:** All relevant data are within the manuscript and its Supporting Information files.

**Funding:** The author(s) received no specific funding for this work.

**Competing interests:** The authors have declared that no competing interests exist.

capacity, equipment, and supplies for regional anesthesia. More engagement of the hospital management, investment in regional anesthesia resources, and training are needed in order to improve the regional anesthesia capacity and provide safe surgery and anesthesia in Botswana.

## Introduction

There is disparity in access to safe surgery and anesthesia between high and low-income countries with an estimate of approximately 5 billion people without access coming mainly from low resources settings (LRS) [1,2]. Regional anesthesia offers multiple advantages especially for LRS such as being low cost, safe, and offering opportunity of expansion of anesthesia practice with labour analgesia and acute pain services [3–6].

Despite known benefits of the regional anesthesia program, many anesthesia providers, and hospitals from LRS face multiple barriers in implementing this program such as lack of enough trainers, lack of equipment and suppliers, and lack of hospital administration support [3–13].

There is a shortage of anaesthesia workforce in Botswana with approximately 76 nurse anaesthetists and 18 anaesthesiologists in public hospitals [14,15]. Anesthesia is mainly provided by nurse anesthetists under minimal supervision. The practice of regional anesthesia in Botswana has been described previously with a focus on spinal anesthesia for caesarean section, however, other types of regional anesthesia procedures have never been explored [16].

Therefore, the aim of this study was to describe the regional anesthesia capacity, characteristics of regional anesthesia practice, and challenges and solutions of practicing safe regional anesthesia in public hospitals in Botswana. The results of this study have potential to inform the design of a regional anesthesia program relevant to the context of public hospitals in Botswana.

## Methods

### Design

This was a cross-sectional survey of anesthesia providers working in public hospitals in Botswana. Reporting followed the Strengthening the Reporting of Observational Studies in Epidemiology (STROBE) guidelines [17].

### Setting of the study

Botswana is a landlocked country in Southern Africa with a landmass of 581 730 $km^2$, a population of 2,588,000, and a life expectancy of approximately 63 years [18]. There are 28 health districts made up of five urban, four rural and 19 rural districts, with one or more urban villages. A shortage of skilled and qualified healthcare workers remains one of the major challenges toward the availability of high-quality healthcare in Botswana with 3.4 doctors and 28.4 nurses per 10 000 people [14,19]. The maternal, under 5 years, and neonatal mortality are still high at 144/100,000 population, 44/100,000, and 22/100,000 population respectively [20]. Due to limited surgical and anaesthesia capacity, surgery is conducted mainly by medical officers and anaesthesia is provided by nurse anaesthetists in most district hospitals across the country [14,21].

Patients still travel long distances or require transfers to get specialized care at a tertiary centre or a private hospital [15,21]. However, there are ongoing efforts from the Ministry of

Health of Botswana to improve the situation by increasing the number of healthcare providers including anaesthesia providers through local training [15]. The anaesthesia residency program at UB is small with 12 residents and only 3 residents are currently rotating in the local hospitals while others are in South Africa for specialized rotations. The nurse anaesthetists training program is in the process to start again after a long period of pause (more than 10 years) due to financial and academic difficulties.

## Population, participants, and data collection process

During the study period, there was a shortage of surgical and anaesthesia workforce in Botswana with 76 nurse anaesthetists, less than 20 clinical officers, and 18 anaesthesiologists [19]. Anaesthesiologists work mainly in two public hospitals (Prince Marina Hospital (PMH) and Francis Town). District hospitals are staffed by medical officers and nurse anaesthetists while primary hospitals are staffed only by nurse anaesthetists. The study questionnaire was adapted from the studies conducted in similar settings in Ethiopia by Merga et, al in 2015 and in Rwanda by Ho et, al in 2019 [12,13]. The research team considered questions relevant to the Botswana context and piloted the final questionnaire to 5 anesthesia providers prior to the start of the study for the understanding (See appendix 1).

A purposive sampling method of public hospitals was used to achieve representation of different hospital levels across the Botswana health system [21] (See Fig 1). All anesthesia providers from selected hospitals were contacted and requested to complete the paper form of the study questionnaire sent to the chief of the anesthesia department by courier. Each participant provided a written consent prior to completing the questionnaire. The research team included all anesthesia providers working in public hospitals in Botswana. Only participants who refused to provide their consent were excluded from the study.

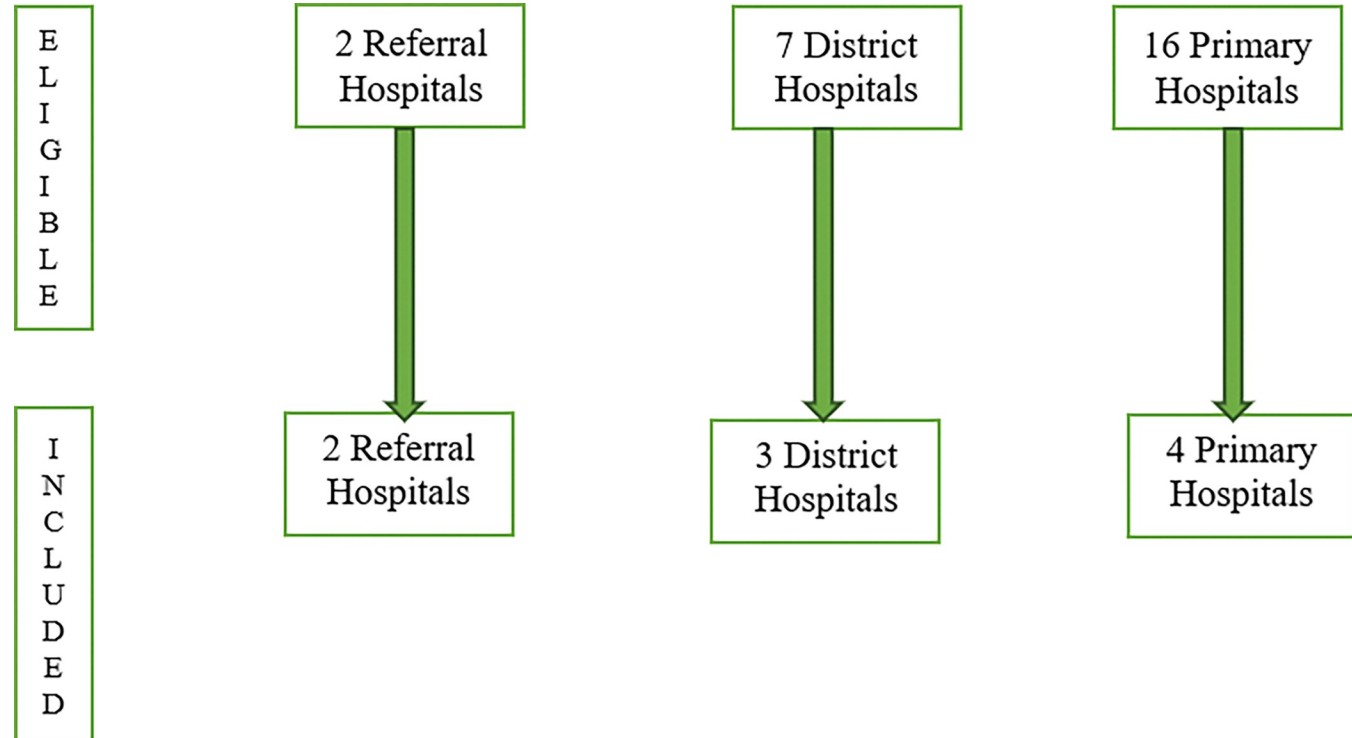

**Fig 1. Flow diagram for selection of study sites.**

To maximize the number of participants, the research team contacted the concerned professionals through each hospital speed dial which is available at PMH. In addition, a list of cell phone/WhatsApp number of all anesthetists available at PMH was used as needed when some professionals were not reached through the speed dial system.

Paper-based questionnaires were distributed to 47 participants. After completion of the questionnaires, the chief of anesthesia department collected all completed questionnaires and sent them to the research team through courier or email. At PMH, the research team collected completed questionnaires directly. A final list of 9 hospitals were retained for the study among 25 hospitals providing anesthesia and eligible for this study (See Fig 1).

### Data analysis

Data were entered electronically into a Microsoft Excel sheet (version 2015) from the paper questionnaires. Descriptive statistics were used to report participants' demographics and characteristics of regional anesthesia practice. Frequencies and percentages were used for categorical data, no continuous data were reported.

### Ethical approval

Ethical approval was obtained from the Institutional Review Board (IRB) at University of Botswana (IRB UBR/RES/IRB/BIO/2021) and Ministry of Health of Botswana (HPDME 13/08/1).

### Results

Forty-seven anesthesia providers received the questionnaires; 38 were returned. This resulted in a response rate of 80.9%.

Most participants were aged between 26 and 45 years (71%), male (65.8%), nurse anesthetists and medical officers (57.8%), from PMH (60.5%), and within 5 years of practice (60.6%) (Table 1).

All 9 hospitals are staffed by medical officers and nurse anesthetists while anesthesiologists are available only in 2 referral hospitals. All 9 hospitals perform spinal anesthesia; however, other regional techniques are performed by a small number of participants from PMH.

All 9 hospitals had adequate regional anesthesia drugs and sedation medications, however, most hospitals (except PMH) lacked ultrasound machine and the regional anesthesia kit (Table 2).

Most participants perform regional anesthesia more than once a week (76.3%), use surface anatomy technique (50%), and use regional anesthesia as sole anesthesia (73.7%). The common surgical indication was gynaecological procedures (76.3%) and common type of training was formal training at the University (Table 3).

The common challenges reported were lack of knowledge and skills, lack of equipment and supplies, and lack of hospital engagement and support. Some solutions were proposed such as regional anesthesia training and engaging the hospital management to get resources (Table 4).

### Discussion

The results of this survey found that spinal anesthesia was the most common regional anesthesia technique in Botswana (performed by every participant) followed by upper limb blocks (supraclavicular block and interscalene block) and epidural anesthesia. Most district and primary hospitals are staffed to provide spinal anesthesia as the only regional anesthesia technique because cesarian section is the only major procedure performed in these settings.

**Table 1. Characteristics of anesthesia providers working in public hospitals in Botswana (N:38).**

| Variable | Number |
|---|---|
| *Qualification* | |
| Anesthesiologists | 4 (10.4) |
| Clinical anaesthetist officer/Diploma | 7 (18.4) |
| BSC anaesthetist | 11 (28.9) |
| Medical officer | 7 (18.4) |
| Anaesthesia residents | 9 (23.7) |
| | |
| *Hospital level* | |
| *1.Referral hospital* | 23 (60.5) |
| Prince Marina Hospital (PMH) | 3 (7.9) |
| Francis Town | 3 (7.9) |
| *2.District hospital* | 2 (5.2) |
| Maun | 3 (7.9) |
| Molopolole | 1 (2.6) |
| Mahalapye | 1 (2.6) |
| *3. Primary hospital* | 1 (2.6) |
| Thamaga | 1 (2.6) |
| Hukuntsi | |
| Ghanzi | |
| Gumare | |
| | |
| *Experience* | |
| <1 year | 5 (13.2) |
| 1–5 years | 18 (47.4) |
| 6–10 years | 3 (7.9) |
| 11–15 years | 1 (2.6) |
| >16–20 years | 4 (10.5) |
| >20 years | 7 (18.4) |

This is similar with findings from multiple studies conducted in low resources settings (LRS) especially in Sub Saharan Africa (SSA) [3,5,6,8,11,13,16,22]. There is lack of enough anesthesiologists in LRS which is further exacerbated with insufficient funding and recognition of anesthesia as a sub-specialty during training [3,5,6,8,11,13,16,22]. Few available anesthesiologists lack the resources to provide training to a critical mass of residents necessary to improve the practice of anesthesiology at the highest level including regional anesthesia as a sub-specialty [3,5,6,8,11,13,16,22]. A well-functioning regional anesthesia program has multiple advantages for LRS because it may improve the pain management capacity, become an alternative anesthesia option for patients who can't tolerate general anesthesia, decrease the patient's cost, and provide new services like labor analgesia for pregnant women [3–5].

There are 3 essential factors which should be considered before establishing a regional anesthesia program in LRS. Training the anesthesia team in knowledge, skills, and attitude is essential along with the organization of the new clinical area for regional anesthesia. In addition, protocols, and guidelines applicable to the local context should be put in place. Furthermore, the required resources (i.e., equipment like ultrasound, supplies like nerve blocks, local anesthesia drugs, and sedative medications) should be put in place with the support from the hospital administration to minimize stock outs and to ensure the sustainability of the program [3–6].

Most participants reported challenges with all the 3 essential requirements to start a well-functioning regional anesthesia program in LRS.

Even if all 3 factors are essential, we would like to recommend investing in training and retaining the initial team of experts which can be done at the level of referral hospitals with the support of the Anesthesia and Critical Care department at the University of Botswana (UB). Once there is an ongoing program, more people from different hospitals may be trained through rotations. For sustainability, the administration support is essential by including regional anesthesia within the anesthesia care expected at the level of referral hospitals.

**Table 2. Regional anesthesia capacity (anesthesia providers, procedures, equipment, supplies, and drugs) for public hospitals in Botswana.**

| Hospital Level | Referral Hospitals | | District Hospitals | | | Primary Hospitals | | | |
|---|---|---|---|---|---|---|---|---|---|
| Hospital Name | PMH | Francis Town | Maun | Molepolole | Mahalapye | Thamaga | Hukntsi | Ghanzi | Gumare |
| **Anesthesia providers (Yes or No)** | | | | | | | | | |
| Anesthesiologist | Yes | Yes | No | No | No | No | No | No | No |
| Residents | Yes | No | No | No | No | No | No | No | No |
| Medical officers | Yes | Yes | Yes | Yes | Yes | Yes | Yes | Yes | Yes |
| Nurse anesthetists | Yes | Yes | Yes | Yes | Yes | Yes | Yes | Yes | Yes |
| **Regional anesthesia procedures (Yes or No)** | | | | | | | | | |
| Spinal Anesthesia | Yes | Yes | Yes | Yes | Yes | Yes | Yes | Yes | Yes |
| Supraclavicular block | Yes | No | No | No | No | No | No | No | No |
| Caudal block | Yes | No | No | No | No | No | No | No | No |
| Wrist block | Yes | No | No | No | No | No | No | No | No |
| Ankle block | Yes | No | No | No | No | No | No | No | No |
| Axillary block | Yes | No | No | No | No | No | No | No | No |
| Transverse abdominus plane block | Yes | No | No | No | No | No | No | No | No |
| Infraclavicular block | Yes | No | No | No | No | No | No | No | No |
| Ilioinguinal | No | No | No | No | No | No | No | No | No |
| Interscalene block | Yes | No | No | No | No | No | No | No | No |
| Combined spinal epidural | **No** | No | No | No | No | No | No | No | No |
| Femoral nerve block | Yes | No | No | No | No | No | No | No | No |
| 3 in 1 block | **No** | No | No | No | No | No | No | No | No |
| Intercostal nerve block | **No** | No | No | No | No | No | No | No | No |
| **Equipment availability (Yes or No)** | | | | | | | | | |
| Nerve stimulator for RA | **No** | No | No | No | No | No | No | No | No |
| Ultrasound for RA | Yes | No | No | No | No | No | No | No | No |
| **Supplies and Drugs availability (Yes or No)** | | | | | | | | | |
| Regional anaesthesia kit | Yes | No | No | No | No | No | No | No | No |
| Regional anaesthesia medications | Yes | Yes | Yes | Yes | Yes | Yes | Yes | Yes | Yes |
| Sedation Drugs | | | | | | | | | |
| Morphine | Yes | Yes | Yes | Yes | Yes | Yes | Yes | Yes | Yes |
| Fentanyl | Yes | Yes | Yes | Yes | Yes | Yes | Yes | Yes | Yes |
| Midazolam | Yes | Yes | Yes | Yes | Yes | Yes | Yes | Yes | Yes |
| Diazepam | Yes | Yes | Yes | Yes | Yes | Yes | Yes | Yes | Yes |

**Table 3. Characteristics of regional anesthesia practice in public hospitals in Botswana, N = 38.**

| Variable | Number (%) |
|---|---|
| *Frequency* | |
| More than once a week | 29 (76.3) |
| Once a week | 6 (15.8) |
| Once a month | 2 (5.3) |
| *Technique* | |
| Nerve stimulator | 2 (5.2) |
| Surface anatomy | 19 (50) |
| Using ultrasound | 14 (36.8) |
| *Indication* | |
| Anaesthesia | 28 (73.7) |
| Post-operative analgesia | 23 (60.5) |
| Analgesia for trauma | 4 (10.5) |
| Less cost | 9 (23.7) |

*(Continued)*

**Table 3.** (Continued)

| Variable | Number (%) |
|---|---|
| *Surgical procedure* | |
| Gynaecological procedures (Hysterectomy, Myomectomy, etc) | 23 (60.5) |
| Caesarean section | 31 (81.6) |
| Orthopaedic procedures | 24 (63.2 |
| General surgery | 21 (55.3) |
| *RA training (other than spinal)* | |
| Short courses | 2 (5.2) |
| Workshops | 0 (0) |
| Formal training at the University | 22 (57.9) |
| *Satisfaction with RA Training* | |
| Very dissatisfied | 10 (26.3) |
| Dissatisfied | 8 (21) |
| Neutral | 6 (15.8) |
| Very satisfied | 6 (15.8) |
| *Use of guidelines* | |
| Guidelines for pain management | 23 (60.5) |
| Guidelines for regional anaesthesia complications management | 20 (52.6) |

For the following variables: Technique, indication, surgical procedure, RA training, use of and guidelines; one participant could choose multiple answers and therefore the total response for each choice/answer is 38 (for each question, the number of responses reflects the number of participants with the experience with the concerned option). For other variables (frequency and satisfaction), participants had to choose one option and therefore the total number of responses is 38 for all options.

Our recommendation of focusing on training first is also supported by the fact that most participants reported lack of knowledge and skills in regional anesthesia even at PMH where most equipment and supplies were available sometimes. Even for hospitals lacking essential equipment like ultrasound and supplies like nerve block needles, with a trained team and a good plan for advocacy the necessary changes and investments can be secured [12,13]. The regional anesthesia program has shown to be cost-effective, to improve patient safety, and to be feasible in LRS [3–6,12,13].

**Table 4. Challenges and solutions of practicing safe regional anesthesia in public hospitals in Botswana.**

| **Major challenges** |
|---|
| *Individual level* |
| Insufficient knowledge of the procedure itself |
| Lack of practical skills to perform the procedure |
| *Organization level* |
| Lack of equipment (i.e. regional procedure kit) |
| Lack of cooperation by operating surgeon to perform the procedure |
| Hospital management is not aware of the importance of RA |
| *Patient level* |
| Patient refusal of the procedure |
| Fear of failure of regional anaesthesia |
| Fear of complications |
| **Proposed solutions** |
| *Individual level* |
| Regional anaesthesia training |
| *Organization level* |
| Involve hospital management to get resources (i.e. ultrasound, nerve stimulator, needles for the block) |
| *Patient level* |
| Teaching sessions using the media for the public to reduce fear and anxiety |

There is paucity of data on current regional anesthesia practice in Botswana and other low-and middle-income countries especially in Sub Saharan Africa. This study is a great contribution to the existing literature on understanding the regional anesthesia practice in LRS towards access to safe anesthesia and surgery. The results highlight the lack of local training capacity, lack of equipment and supplies, and inadequate administration support as major barriers to the delivery of regional anesthesia services in Botswana.

This article can be used for advocacy for more investments for safe regional anesthesia programs in Botswana and other LRS.

There are limitations to consider while interpreting the results of this study. First, the sample size is small and include only a few hospitals, the results may not be generalizable to all hospitals in Botswana. Second, there were few anesthesiologists among participants who completed the survey, this might have led to under consideration of the capacity of conducting different types of regional anesthesia procedures. Third, this study included only participants working in public hospitals and doesn't provide a picture of the current regional anesthesia practice in private hospitals in Botswana which are usually staffed by anesthesiologists trained from South Africa, China, Cuba, Ethiopia, and India.

However, if there is more regional anaesthesia being performed in private hospitals, it means the capacity to train staff in RA is available locally and can be harnessed through the Botswana Society of Anaesthesiologists.

## Conclusion

The results of this study suggest that spinal anesthesia is the most common regional anesthesia technique performed by anesthesia providers working in public hospitals in Botswana followed by few upper limb blocks. However, most public hospitals lack enough training capacity, equipment, and supplies for regional anesthesia. More engagement of the hospital management, investment in regional anesthesia resources, and training are needed in order to improve the regional anesthesia capacity and provide safe surgery and anesthesia in Botswana. Further studies should explore barriers and facilitators to the implementation of a new regional anesthesia program in public hospitals in Botswana.

## Acknowledgments

The authors thank the chiefs of anesthesia departments in different study hospitals for their support to the research team during the whole project especially with distribution and return of questionnaires.

## Author Contributions

**Conceptualization:** Mamo Kassa, Farai Madzimbamuto, Gaone Kediegite.

**Data curation:** Mamo Kassa.

**Formal analysis:** Mamo Kassa, Eugene Tuyishime.

**Methodology:** Mamo Kassa, Farai Madzimbamuto, Gaone Kediegite.

**Project administration:** Mamo Kassa.

**Writing – original draft:** Mamo Kassa, Farai Madzimbamuto, Eugene Tuyishime.

**Writing – review & editing:** Mamo Kassa, Farai Madzimbamuto, Gaone Kediegite, Eugene Tuyishime.

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
