## [Decision Letter · Decision Letter 0]

29 Jun 2023

PONE-D-23-09813Regional anaesthesia practice in public hospitals in Botswana: A cross-sectional studyPLOS ONE

Dear Dr. Tuyishime,

Thank you for submitting your manuscript to PLOS ONE. After careful consideration, we feel that it has merit but does not fully meet PLOS ONE’s publication criteria as it currently stands. Therefore, we invite you to submit a revised version of the manuscript that addresses the points raised during the review process.

ACADEMIC EDITOR: Please revise the manuscript according to the reviewers' suggestions. 

We look forward to receiving your revised manuscript.

Kind regards,

Silvia Fiorelli

Academic Editor

PLOS ONE

Journal Requirements:

3. We note that Figure 1 in your submission contain map images which may be copyrighted. All PLOS content is published under the Creative Commons Attribution License (CC BY 4.0), which means that the manuscript, images, and Supporting Information files will be freely available online, and any third party is permitted to access, download, copy, distribute, and use these materials in any way, even commercially, with proper attribution. For these reasons, we cannot publish previously copyrighted maps or satellite images created using proprietary data, such as Google software (Google Maps, Street View, and Earth). For more information, see our copyright guidelines: http://journals.plos.org/plosone/s/licenses-and-copyright.

Reviewers' comments:

Reviewer's Responses to Questions

**Comments to the Author**

1. Is the manuscript technically sound, and do the data support the conclusions?

Reviewer #1: Partly

Reviewer #2: Yes

2. Has the statistical analysis been performed appropriately and rigorously? 

Reviewer #1: No

Reviewer #2: No

3. Have the authors made all data underlying the findings in their manuscript fully available?

Reviewer #1: Yes

Reviewer #2: Yes

4. Is the manuscript presented in an intelligible fashion and written in standard English?

Reviewer #1: No

Reviewer #2: Yes

5. Review Comments to the Author

Reviewer #1: Thank you for inviting me to review this manuscript, which presents the results of a study to describe the regional anesthesia capacity, characteristics of regional anesthesia practice, and challenges and solutions to practicing safe regional anesthesia in public hospitals in Botswana.

The study is on an interesting topic, of broad interest to the readership and I applaud the authors for undertaking this study, and their attempts to raise awareness about regional anesthesia practices.

General comments:

The work is overall clearly presented and easy to read. In my opinion, the major weakness of the study relates to the methods; participant selection, data collection, and analysis

Specific comments:

Keywords- it is better to add Botswana, than repetitions

Competing interest – delete he/the---

ES. Line 1 - resources settings

Purposive sampling-not clearly stated and implemented in the methods section.

approximately 76 nurse anesthetists and 18 anesthesiologists in public hospitals- the figure is since 2021 but there has been an ongoing residency program

All anesthesia providers were contacted and requested to complete the paper form of the study questionnaire sent to the chief of the anesthesia department by courier. Why do you need to use a courier when there are electronic data collection options?

A purposive sampling method of public hospitals was used to achieve representation of different hospital levels across the Botswana health system- however, most of the participant was from Prince Marina Hospital (PMH)- 23 (60.5).

Result section

Table 1. N=38

It is good to mention the numerical value and then the percentage in a bracket

Age total=37 where is the remaining 1

Prince Marina Hospital (PMH)- 23 (60.5) of the study participant were from a single hospital and how the figures in this study will be inferred from the practice of regional anesthesia in Botswana

What is the essence of investigating the sociodemographic status of anesthesia providers unless it has a statistical analysis in association with the practice

Table 2. Regional anesthesia capacity (anesthesia providers, procedures, equipment, supplies, and drugs) for public hospitals in Botswana

How do we know the specific practice is not performed either due to a lack of knowledge and skills or basic essential equipment for each regional anesthesia technique?

Table 3: N= 38

Regarding the technique of block, what do “N-37” and “%” implies? A single anesthesia provider can use three of the approaches mentioned and the figure depicted in the table needs elaboration. The same is true for the rest of the sub-topics in the table.

RA-training- Only 24 (63.1) have formal training at the University or a short course in regional anesthesia but all the study participants can at least perform spinal anesthesia. Is that possible to practice without any training, how is the standard of practice negotiated here?

Table 4:

It would have been good thematically analyze the challenges in terms of academic qualification, institution, and year of experience … with its mitigation strategies

Discussion and conclusions:

Those are elaborations of the whole study with other similar studies and the findings. Hence, the discussion shall be revised based on the feedback raised in the methods and result section.

It needs to be revised and needs to be very clear about what this adds to the existing literature and clearly details learning points.

Overall needs a further major revision of the methodology, English grammar, and syntax to write in a scientific voice fashion and the journal protocols.

Reviewer #2: Dear authors, here you receive my remarks concerning the manuscript entitled

“Regional anaesthesia practice in public hospitals in Botswana: A cross-sectional study” and

Short Title: Regional anaesthesia practice in public hospitals in Botswana with number PONE-D-23-09813.

The authors present a observational study and show the results of cross-sectional survey among anesthesia providers working in public hospitals in Botswana. The article is well written and provides a clear cut idea and simple statistics. A number approval of the medical ethical board is provided. I think that it is an interesting manuscript and important for the situation in Botswana concerning the daily practice in regional anesthesia.

I have a few remarks:

Abstract: (Results)… by a small number of participants one referral hospital. Please change into:…by a small number of participants in one referral hospital.

P5 Setting of the study. Is it possible to provide some more information about the greater perspective, i.e. how many anesthesia providers are there in total? You mention about 76 nurse anesthetists and 18 anesthesiologists. Am I correct that you received only from 4 anesthesiologists an answer concerning the survey? How many hospitals are there actually? You mention 25 (Figure 2)! It would improve understanding to actually relate your results in the greater perspective of the total and in what way it is the best representing sample provided here?

P13 Can you tell us in how many of the cases locoregional anesthesia was used in relation of the number that could have been performed and was chosen to perform general anesthesia due to the reasons mentioned in Table 4, with of course the exclusion of the patients who did not choose locoregional anesthesia? In other words, is it possible for instance to calculate how many women in labour received general anesthesia and could have received spinal anesthesia but due to the reasons presented in Table 4 this did not happen. This is only easy one example, there may be more.

Regarding Table 4: Did you become aware of shortage also to treat possible Local anesthetic systemic toxicity (LAST). Any idea about the incidence and the outcome of this serious complication?

Textual: Do you mean “limited resources settings" or limited resource settings? Please compare when searching in the internet?

Reference double! Ref 13= ref23

6. PLOS authors have the option to publish the peer review history of their article (what does this mean?). If published, this will include your full peer review and any attached files.

Reviewer #1: No

Reviewer #2: **Yes: **P. Bruins

---

## [Author Response · Author response to Decision Letter 0]

5 Jul 2023

Eugene TUYISHIME, MBBS, MSc, 05th July, 2023

Anesthesiologist

Assistant Program Director, 

Anesthesia and Critical Care department

University of Botswana, 

Phone: + 26774504984

Email: tuyishime36@gmail.com

To: Editor of Plos One Journal

Dear Editor,

RE: Re-submission of the manuscript/PONE-D-23-09813 entitled “Regional anaesthesia practice in public hospitals in Botswana: A cross-sectional study”

I am pleased to submit a revised version of our study to Plos One journal.

Our responses to your comments and the reviewer’s questions are highlighted in text in blue both in this letter and the manuscript text.

Sincerely,

Dr Eugene Tuyishime

Editor and Reviewer Comments:   

Academic Editor (Silvia Fiorelli)

Journal Requirements:

https://journals.plos.org/plosone/s/file?id=wjVg/PLOSOne_formatting_sample_main_body.pdfand

We have addressed the style requirements as suggested.

We have provided details about informed consent in the data collection process section as follows:

. “Each participant provided a written consent prior to completing the questionnaire.”

2. We note that Figure 1 in your submission contain map images which may be copyrighted. All PLOS content is published under the Creative Commons Attribution License (CC BY 4.0), which means that the manuscript, images, and Supporting Information files will be freely available online, and any third party is permitted to access, download, copy, distribute, and use these materials in any way, even commercially, with proper attribution. For these reasons, we cannot publish previously copyrighted maps or satellite images created using proprietary data, such as Google software (Google Maps, Street View, and Earth). For more information, see our copyright guidelines: http://journals.plos.org/plosone/s/licenses-and-copyright.

We have removed Figure 1 from our re-submission.

We have not submitted any extra files.

Reviewer #1: 

Thank you for inviting me to review this manuscript, which presents the results of a study to describe the regional anesthesia capacity, characteristics of regional anesthesia practice, and challenges and solutions to practicing safe regional anesthesia in public hospitals in Botswana.

The study is on an interesting topic, of broad interest to the readership and I applaud the authors for undertaking this study, and their attempts to raise awareness about regional anesthesia practices.

Thank you for your feedback.

General comments:

The work is overall clearly presented and easy to read. In my opinion, the major weakness of the study relates to the methods; participant selection, data collection, and analysis

Specific comments:

Keywords- it is better to add Botswana, than repetitions

We have added Botswana and removed repetitions (Low Resource settings and Sub Saharan Africa) as suggested.

Competing interest – delete he/the---

Deleted as suggested.

ES. Line 1 - resources settings

Purposive sampling-not clearly stated and implemented in the methods section.

approximately 76 nurse anesthetists and 18 anesthesiologists in public hospitals- the figure is since 2021 but there has been an ongoing residency program

All anesthesia providers were contacted and requested to complete the paper form of the study questionnaire sent to the chief of the anesthesia department by courier. Why do you need to use a courier when there are electronic data collection options?

We have rephrased this section to make it clear, as follows:

“All anesthesia providers from selected hospitals were contacted and requested to complete the paper form of the study questionnaire sent to the chief of the anesthesia department by courier.”

We decided to use courier and paper forms to minimize the risk of low response rate due to lack of familiarity in completing electronic survey or excluding some participants without access to internet.

A purposive sampling method of public hospitals was used to achieve representation of different hospital levels across the Botswana health system- however, most of the participant was from Prince Marina Hospital (PMH)- 23 (60.5).

Many participants came from PMH because it is the major hospital with a bigger number of anesthesia providers. Also, the research team works at PMH and had more access to participants which may have led to higher response rate. However, we believe that the data collected across all sites provides an appropriate representation of the current practice of regional anesthesia in public hospital in Botswana.

Result section

Table 1. N=38

It is good to mention the numerical value and then the percentage in a bracket

Age total=37 where is the remaining 1

There is a missing information from one participant. However, we have removed this information from the table as suggested below.

Prince Marina Hospital (PMH)- 23 (60.5) of the study participant were from a single hospital and how the figures in this study will be inferred from the practice of regional anesthesia in Botswana.

PMH has more anesthesia providers as the major hospital in Botswana. Other smaller hospitals had enough representation based on their staffing capacity which could have been as low as one or two anesthesia providers. We believe that the low number of participants from smaller hospitals is an issue of shortage of staff of those hospitals and not a problem of low response rate; therefore, we can trust the results to represent the current practice in those hospitals. It is also an opportunity to highlight a severe shortage of anesthesia providers in smaller hospitals in Botswana.

What is the essence of investigating the sociodemographic status of anesthesia providers unless it has a statistical analysis in association with the practice

We have removed this information from the table as suggested. We have kept 3 characteristics of participants relevant to regional anesthesia practice (qualification, Hospital level, and experience).

Table 2. Regional anesthesia capacity (anesthesia providers, procedures, equipment, supplies, and drugs) for public hospitals in Botswana

How do we know the specific practice is not performed either due to a lack of knowledge and skills or basic essential equipment for each regional anesthesia technique?

Thank you for your comment. The purpose of this table was to describe the current situation in practicing regional anesthesia in public hospitals in Botswana. The reasons of not practicing some techniques were addressed in Table 4.

Table 3: N= 38

Regarding the technique of block, what do “N-37” and “%” implies? A single anesthesia provider can use three of the approaches mentioned and the figure depicted in the table needs elaboration. The same is true for the rest of the sub-topics in the table.

We have added some explanation below the table as follows:

For the following variables: technique, indication, surgical procedure, RA training, use of and guidelines; one participant could choose multiple answers and therefore the total response for each choice/answer is 38 (for each question, the number of responses reflects the number of participants with the experience with the concerned option). For other variables (frequency and satisfaction), participants had to choose one option and therefore the total number of responses is 38 for all options.

RA-training- Only 24 (63.1) have formal training at the University or a short course in regional anesthesia but all the study participants can at least perform spinal anesthesia. Is that possible to practice without any training, how is the standard of practice negotiated here?

We have corrected the variable for more clarification as follows:

RA training (other than spinal).

The question was about regional anaesthesia training excluding spinal anaesthesia because every anaesthesia provider in Botswana had good training in performing spinal anaesthesia. However, other regional anaesthesia techniques are not taught either because of lack of resources and trainers (for physician anaesthesiologists) or because they are not part of the curriculum and scope of practice for nurse anaesthetists.

Table 4:

It would have been good thematically analyze the challenges in terms of academic qualification, institution, and year of experience … with its mitigation strategies

Thank you for your comment. As this question was asked as free text, we tried to capture common answers which were given by all groups of participants. We have modified the table to provide a systematic analysis at 3 levels (individual, organization, and patient).

Discussion and conclusions:

Those are elaborations of the whole study with other similar studies and the findings. Hence, the discussion shall be revised based on the feedback raised in the methods and result section.

It needs to be revised and needs to be very clear about what this adds to the existing literature and clearly details learning points.

Thank you for your comment. We have added a paragraph to explain what this manuscript adds to the current literature, as follows:

“There is paucity of data on current regional anesthesia practice in Botswana and other low- and middle-income countries especially in Sub Saharan Africa. This study is a great contribution to the existing literature on understanding the regional anesthesia practice in LRS towards access to safe anesthesia and surgery. The results highlight the lack of local training capacity, lack of equipment and supplies, and inadequate administration support as major barriers to the delivery of regional anesthesia services in Botswana. This article can be used for advocacy for more investments for safe regional anesthesia programs in Botswana and other LRS.”

Overall needs a further major revision of the methodology, English grammar, and syntax to write in a scientific voice fashion and the journal protocols.

Thank you for your feedback. We have improved the methodology and scientific writing as suggested. We believe that our manuscript has been improved to meet the publication requirements of Plos One.

Reviewer #2: 

Dear authors, here you receive my remarks concerning the manuscript entitled

“Regional anaesthesia practice in public hospitals in Botswana: A cross-sectional study” and

Short Title: Regional anaesthesia practice in public hospitals in Botswana with number PONE-D-23-09813.

The authors present an observational study and show the results of cross-sectional survey among anesthesia providers working in public hospitals in Botswana. The article is well written and provides a clear cut idea and simple statistics. A number approval of the medical ethical board is provided. I think that it is an interesting manuscript and important for the situation in Botswana concerning the daily practice in regional anesthesia.

Thank you for your feedback.

I have a few remarks:

Abstract: (Results)… by a small number of participants one referral hospital. Please change into:…by a small number of participants in one referral hospital.

We have changed this as suggested.

P5 Setting of the study. Is it possible to provide some more information about the greater perspective, i.e. how many anesthesia providers are there in total? You mention about 76 nurse anesthetists and 18 anesthesiologists. Am I correct that you received only from 4 anesthesiologists an answer concerning the survey? How many hospitals are there actually? You mention 25 (Figure 2)! It would improve understanding to actually relate your results in the greater perspective of the total and in what way it is the best representing sample provided here?

We have added more details to provide greater perspective as suggested:

During the study period, there was a shortage of surgical and anaesthesia workforce in Botswana with 76 nurse anaesthetists, less than 20 clinical officers, and 18 anaesthesiologists19. Anaesthesiologists work mainly in two public hospitals (Prince Marina Hospital (PMH) and Francis Town). District hospitals are staffed by medical officers and nurse anaesthetists while primary hospitals are staffed only by nurse anaesthetists.

P13 Can you tell us in how many of the cases locoregional anesthesia was used in relation of the number that could have been performed and was chosen to perform general anesthesia due to the reasons mentioned in Table 4, with of course the exclusion of the patients who did not choose locoregional anesthesia? In other words, is it possible for instance to calculate how many women in labour received general anesthesia and could have received spinal anesthesia but due to the reasons presented in Table 4 this did not happen. This is only easy one example, there may be more.

Thank you for your comment. We don’t have this information because the main objective of the study was to evaluate the capacity of the anesthesia providers and their hospitals in delivering regional anesthesia services without getting more details on the actual care of individual patients. This may be addressed by follow up studies. However, there is a previous study done by Kassa and colleagues in 2017, on types of anesthesia for cesarean section (CS) in Botswana, which showed that most CS were done under spinal anesthesia (at a rate of 95.2%) (see this article: Kassa MW, Mkubwa JJ, Shifa JZ, Agizew TB. Type of anaesthesia for caesarean section and failure rate in Princess Marina hospital, Botswana’s largest referral hospital. Afr Health Sci.2020;20(3):1229-1236. doi:10.4314/ahs.v20i3.26).

Regarding Table 4: Did you become aware of shortage also to treat possible Local anesthetic systemic toxicity (LAST). Any idea about the incidence and the outcome of this serious complication?

Thank you for your comment. We don’t have this information. We corrected data on the availability of the protocols to manage LAST. The incidence of LAST can be addressed by follow up studies.

Textual: Do you mean “limited resources settings" or limited resource settings? Please compare when searching in the internet?

We have corrected this to low resource settings after checking in the literature.

Reference double! Ref 13= ref23

We have corrected the reference as suggested and kept reference 13.

---

## [Decision Letter · Decision Letter 1]

23 Oct 2023

PONE-D-23-09813R1Regional anaesthesia practice in public hospitals in Botswana: A cross-sectional studyPLOS ONE

Dear Dr. Tuyishime,

Thank you for submitting your manuscript to PLOS ONE. After careful consideration, we feel that it has merit but does not fully meet PLOS ONE’s publication criteria as it currently stands. Therefore, we invite you to submit a revised version of the manuscript that addresses the points raised during the review process.

ACADEMIC EDITOR: Please carefully assess all reviewer's comments==============================

We look forward to receiving your revised manuscript.

Kind regards,

Silvia Fiorelli

Academic Editor

PLOS ONE

Journal Requirements:

Reviewers' comments:

Reviewer's Responses to Questions

**Comments to the Author**

1. If the authors have adequately addressed your comments raised in a previous round of review and you feel that this manuscript is now acceptable for publication, you may indicate that here to bypass the “Comments to the Author” section, enter your conflict of interest statement in the “Confidential to Editor” section, and submit your "Accept" recommendation.

Reviewer #1: All comments have been addressed

Reviewer #2: All comments have been addressed

2. Is the manuscript technically sound, and do the data support the conclusions?

Reviewer #1: Yes

Reviewer #2: Yes

3. Has the statistical analysis been performed appropriately and rigorously? 

Reviewer #1: Yes

Reviewer #2: N/A

4. Have the authors made all data underlying the findings in their manuscript fully available?

Reviewer #1: Yes

Reviewer #2: Yes

5. Is the manuscript presented in an intelligible fashion and written in standard English?

Reviewer #1: Yes

Reviewer #2: Yes

6. Review Comments to the Author

Reviewer #1: Sorry for the inconveniences and delay in the publication process.

Despite limitations on methodological robustness the authors tried their best to incorporate the given feedback, the research

work is imperative, informative and gives insight for policymakers ,funding agencies and professional development .

Hence, I suggest the acceptance of the manuscript with minor revisions.

Reviewer #2: Dear authors,

here you receive my 2nd review regarding the manuscript with the title: Regional anaesthesia practice in public hospitals in Botswana: A cross-sectional study (Regional anaesthesia practice in public hospitals in Botswana) and Manuscript Number: PONE-D-23-09813R1.

The authors have carefully edited the manuscript and taken our previous criticisms seriously to improve understanding and clarity of the article and have answered our questions accordingly.

I only have one text comment. Please look at the Plos One’s Style requirements more carefully and accurately as requested by the Academic Editor (Silvia Fiorelli). There should be uniformity in the way references are numbered in the text, ie with or without a space after the last letter of a word, with or without brackets, or as a superscript. See examples on the following pages:

page 5: … guidelines(17). And more examples on this page and further on this page and in the manuscript

page 6: … hospital(15,21) and line 4: …training 15 and in the discussion on P13 : … women 3-5

and page 14: .. program (3-6) or …LRS (3-6, 12,13)

Please check other places within the article also?

7. PLOS authors have the option to publish the peer review history of their article (what does this mean?). If published, this will include your full peer review and any attached files.

Reviewer #1: No

Reviewer #2: **Yes: **p.bruins

---

## [Author Response · Author response to Decision Letter 1]

28 Oct 2023

Eugene TUYISHIME, MBBS, MSc, 28th October, 2023

Anesthesiologist

Assistant Program Director, 

Anesthesia and Critical Care department

University of Botswana, 

Phone: + 26774504984

Email: tuyishime36@gmail.com

To: Editor of Plos One Journal

Dear Editor, Silvia Fiorelli,

RE: Re-submission of the manuscript/PONE-D-23-09813 entitled “Regional anaesthesia practice in public hospitals in Botswana: A cross-sectional study”

I am pleased to submit a revised version of our study to Plos One journal.

Our responses to your comments and the reviewer’s questions are highlighted in text in blue both in this letter and the manuscript text.

Sincerely,

Dr Eugene Tuyishime

Editor and Reviewer Comments:   

Academic Editor (Silvia Fiorelli)

Journal Requirements:

We have addressed the style requirements as suggested.

Reviewer #1: 

Sorry for the inconveniences and delay in the publication process.

Despite limitations on methodological robustness the authors tried their best to incorporate the given feedback, the research work is imperative, informative and gives insight for policymakers, funding agencies and professional development.

Hence, I suggest the acceptance of the manuscript with minor revisions.

Thank you for your feedback and suggestions throughout the review process.

Reviewer #2: 

Dear authors,

Here you receive my 2nd review regarding the manuscript with the title: Regional anaesthesia practice in public hospitals in Botswana: A cross-sectional study (Regional anaesthesia practice in public hospitals in Botswana) and Manuscript Number: PONE-D-23-09813R1.

The authors have carefully edited the manuscript and taken our previous criticisms seriously to improve understanding and clarity of the article and have answered our questions accordingly.

I only have one text comment. Please look at the Plos One’s Style requirements more carefully and accurately as requested by the Academic Editor (Silvia Fiorelli). 

There should be uniformity in the way references are numbered in the text, ie with or without a space after the last letter of a word, with or without brackets, or as a superscript. 

See examples on the following pages:

page 5: … guidelines(17). And more examples on this page and further on this page and in the manuscript

page 6: … hospital(15,21) and line 4: …training 15 and in the discussion on P13 : … women 3-5

and page 14: .. program (3-6) or …LRS (3-6, 12,13)

Please check other places within the article also?

Thank you for your feedback. We have addressed the style requirements throughout the manuscript as suggested.

---

## [Decision Letter · Decision Letter 2]

4 Dec 2023

Regional anaesthesia practice in public hospitals in Botswana: A cross-sectional study

PONE-D-23-09813R2

Dear Dr. Tuyishime,

We’re pleased to inform you that your manuscript has been judged scientifically suitable for publication and will be formally accepted for publication once it meets all outstanding technical requirements.

Kind regards,

Silvia Fiorelli

Academic Editor

PLOS ONE

Additional Editor Comments (optional):

Congratulations to the authors and thanks to the reviewers for the provided suggestions which really helped improve the quality of the manuscript

Reviewers' comments:

Reviewer's Responses to Questions

**Comments to the Author**

1. If the authors have adequately addressed your comments raised in a previous round of review and you feel that this manuscript is now acceptable for publication, you may indicate that here to bypass the “Comments to the Author” section, enter your conflict of interest statement in the “Confidential to Editor” section, and submit your "Accept" recommendation.

Reviewer #1: All comments have been addressed

Reviewer #2: All comments have been addressed

2. Is the manuscript technically sound, and do the data support the conclusions?

Reviewer #1: Yes

Reviewer #2: Yes

3. Has the statistical analysis been performed appropriately and rigorously? 

Reviewer #1: Yes

Reviewer #2: Yes

4. Have the authors made all data underlying the findings in their manuscript fully available?

Reviewer #1: Yes

Reviewer #2: Yes

5. Is the manuscript presented in an intelligible fashion and written in standard English?

Reviewer #1: Yes

Reviewer #2: Yes

6. Review Comments to the Author

Reviewer #1: The Authors revised all the feedbacks meticulously. Hence, from my perspective, it is reasonable to give green light for full consideration of publication.

Reviewer #2: Dear authors,

here you receive my 3 rd review regarding the manuscript with the title: Regional anaesthesia practice in public hospitals in Botswana: A cross-sectional study (Regional anaesthesia practice in public hospitals in Botswana) and Manuscript Number: PONE-D-23-09813R2.

The authors have again carefully edited the manuscript and taken our previous criticisms seriously to improve understanding and clarity of the article and have answered our questions accordingly. Finally, The authors have taken the criticism to heart and adapted the texts and references to the proposed Plos one’s house style. I have no further comment.

7. PLOS authors have the option to publish the peer review history of their article (what does this mean?). If published, this will include your full peer review and any attached files.

Reviewer #1: No

Reviewer #2: No

---

## [Editor Report · Acceptance letter]

7 Dec 2023

PONE-D-23-09813R2 

Regional anaesthesia practice in public hospitals in Botswana: A cross-sectional study 

Dear Dr. Tuyishime:

I'm pleased to inform you that your manuscript has been deemed suitable for publication in PLOS ONE. Congratulations! Your manuscript is now with our production department. 

Kind regards, 

on behalf of

Dr. Silvia Fiorelli 

Academic Editor

PLOS ONE